# ViSt3D: Video Stylization with 3D CNN

**Ayush Pande**
IIT Kanpur
ayushp@cse.iitk.ac.in

**Gaurav Sharma**
TensorTour & IIT Kanpur
gaurav@tensortour.com

## Abstract

Visual stylization has been a very popular research area in recent times. While image stylization has seen a rapid advancement in the recent past, video stylization, while being more challenging, is relatively less explored. The immediate method of stylizing videos by stylizing each frame independently has been tried with some success. To the best of our knowledge, we present the first approach to video stylization using 3D CNN directly, building upon insights from 2D image stylization. Stylizing video is highly challenging, as the appearance and video motion, which includes both camera and subject motions, are inherently entangled in the representations learnt by a 3D CNN. Hence, a naive extension of 2D CNN stylization methods to 3D CNN does not work. To perform stylization with 3D CNN, we propose to explicitly disentangle motion and appearance, stylize the appearance part, and then add back the motion component and decode the final stylized video. In addition, we propose a dataset, curated from existing datasets, to train video stylization networks. We also provide an independently collected test set to study the generalization of video stylization methods. We provide results on this test dataset comparing the proposed method with 2D stylization methods applied frame by frame. We show successful stylization with 3D CNN for the first time, and obtain better stylization in terms of texture cf. the existing 2D methods. Project page: https://ayush202.github.io/projects/ViSt3D.html

## 1   Introduction

Image stylization is the task of transforming an input *content* image into a stylized image with the style taken from another input *style* image. It has been a very popular research topic as well as a widely used image editing tool in the recent past. Pictures taken by many users have been transformed into the styles of Picasso and Monet. While image stylization with 2D CNNs has been hugely popular [4, 9, 11, 16], stylization of videos has been relatively less explored. Some recent works on image stylization also show results on video stylization by doing frame by frame image stylization, e.g. [4, 11, 18], and have achieved some success. 3D CNN architectures designed for video processing, to the best of our knowledge, have not yet been made amenable for video stylization at all. Since the 3D convolutions used in the 3D CNNs, inherently combine the spatial and temporal information, the two factors of variations are deeply entangled in the resulting representations. This makes video stylization with 3D CNNs extremely challenging as the style is usually for the appearance, i.e. indicated by a single image, and modifying the video representation to incorporate style in the content, but not in motion is a very challenging task. Naive extension of 2D stylization, e.g. directly transferring feature statistics from style to content features using AdaIN [11] operation with 3D CNNs intermediate features, leads to unusable video output, with intermittent slow motion and jerky transitions due to the style input being static (see supplementary material for representative results).

We address the challenging task of video stylization using 3D CNNs directly. We propose an architecture and training method which takes a content video clip and a style image, and produces an aesthetically pleasing stylized video clip. We do so by disentangling the appearance and motion

37th Conference on Neural Information Processing Systems (NeurIPS 2023).

features from the encoder network and applying AdaIN operation extended for 3D CNN features, coupled with feature map reconstruction loss and style statistics losses. We show how to directly extend AdaIN operation to work with 3D CNN based encoders and decoders, which we call AdaIN3D, and train the network successfully despite the challenges with highly entangled appearance and motion information. All the sub-networks used in the proposed architecture are based on 3D CNN. Since a dataset was not available for the task of studying video stylization, we also propose a large-scale dataset with 10, 000 content clips curated from public benchmark Sports1M [14], paired with the train set of style images from WikiArt dataset [20] which are used to train 2D stylization methods. We show qualitatively that the proposed method produces better results than frame by frame stylization using state-of-the-art 2D image stylization methods. We also give quantitative results based on optical flow errors, comparing the results of the proposed method to that of the state-of-the-art 2D stylization methods. Our contributions can be summarized as follows.

- We propose a novel method for video stylization using 3D CNN, and we are the first to report that stylization with 3D CNN is possible.
- We propose a novel approach to disentangle the appearance and motion features, which are deeply entangled in representations learnt by 3D CNN networks like C3D [25].
- We curated the video dataset from the public Sports1M dataset [14] by filtering videos with motion automatically and use it for training our video stylization network.
- We use existing loss functions and design new ones to encourage learning of artifact free video stylization.
- We provide exhaustive comparisons with existing methods, detailing the differences.

Overall, since visual stylization is effectively a subjective creative filter, we hope to add one more tool to the video creatives toolbox.

## 2   Related works

**Image stylization.**   Gatys et al. [8] proposed an approach to generate textures using CNN and optimization based technique. They used the loss function over Gram matrices on feature maps for maintaining feature map statistics and hence generating required textures. In their next work, Gatys et al. [9] used a similar optimization based approach for stylizing content image with the texture of a style image. Such optimization based approach is quite slow, and the training has to be performed individually for every new pair of images. The next generation of image stylization approaches then turned to feedforward CNNs, e.g. [13, 26], for speed and avoiding retraining on every test style image. However, they still could not be used with arbitrary style images on the fly, i.e. one network needs to be trained for each style image. Finally, the current generation of image stylization methods emerged that were designed for universal image stylization [11, 17, 28], i.e. the network is trained once, and then is used for transferring style from an arbitrary style image, fed along with the content image as input at test time. AdaIN [11] was a very successful step towards universal style transfer because of the simplicity of the network training and parameter free transfer of content into the style space using the AdaIN operation. Among other works, Li et al. [17] used whitening and color transforms for style transfer. However, such transformations produce unsatisfactory results, potentially loosing/modifying content in the output image. SANet [19] tried to introduce more style into the network but traded off the content from the original content image.

**Video stylization.**   All the image stylization methods can be applied to videos by processing the videos frame by frame. Video stylization when done with image stylization methods such as AdaIN [11], WCT [17] etc., leads to stylized videos which suffer from various temporal and spatial inconsistencies. Chen et al. [2] and Gao et al. [7] hence proposed temporal consistency constraints to address these. Methods for video stylization include MCCNet [3] and AdaAttn [18]. MCCNet [3] mainly focused on video stylization, while AdaAttn [18] addressed image stylization using attention and extended the method for videos as well. Both methods lead to stylized video with some degree of style transfer, but increasing the style transfer introduces artifacts quickly. We compare with both of these in our experiments. Wang et al. [27] added optical flow to improve video stylization with 2D CNNs, and here we also use optical flow-based loss but with 3D CNN directly. We observed that there is noticeable flickering in some examples generated by Wang et al. [27]. Also, it smooths out

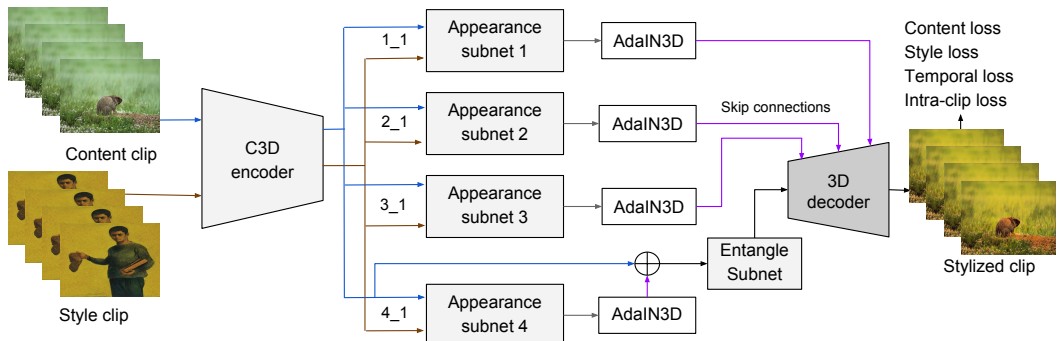

Figure 1: Full architecture for the proposed network for video stylization. The network consists of 3D CNN based encoder and decoder networks, along with two other critical parts: (i) appearance subnets and (ii) Entangle layer. The appearance subnets are trained to disentangle appearance and motion components from the full 3D CNN feature and extract out the appearance part only. The entangle layer is trained to do the opposite, i.e. given the (stylized) appearance features as well as the full 3D CNN features, it re-introduces the motion component in the stylized features to enable them to be decoded into a stylized video by the decoder. The contribution of this paper is to report successful video stylization with 3D CNN directly, cf. 2D CNNs of existing works, using the above network, and a multi-phase training method to train the individual sub-networks proposed.

the low-level textural details, while the proposed method does not have heavy flickering, and it also maintains the low-level textural details.

**Stylization with 2D and 3D CNNs.** All the above methods mainly work with 2D CNNs, e.g. [10, 15, 21, 24], for encoding the content and style images, and propose methods for transferring style by matching statistics of the intermediate feature maps. Videos are inherently 3D, and 3D CNNs, e.g. [1, 25], have been proposed for learning with videos. There has been no work on video stylization with 3D CNN, and in this paper we report the first successful network and training method to do stylization with 3D CNN. Other than breaking new ground, we also show that the proposed method produces richer texture in the stylized video, and since stylization is contextual and subjective, it is at the very least, another useful tool for video creatives.

## 3 Approach

We now present the proposed network architecture and training method for doing video stylization with 3D CNNs. Since we need to perform the challenging task of disentangling the motion and appearance components, the method is trained in multiple phases. In the following, we first describe the full architecture used at inference time in Section 3.1, which takes the content video clip as well as the style clip (the style image repeated $K$ times for clips of size $K$), giving details of each subnetwork. Then, we explain the losses and the multi-phase training method in Section 3.3.

### 3.1 Overall network

The task addressed here is to do video stylization with a reference style image. The network takes as input the content clip (which needs to be stylized) and the reference style image, and outputs the corresponding stylized video.

Figure 1 shows the full network used for stylizing input video clip, with an input style image. Along with the input video clip, the network takes a style clip as input. The style clip is constructed by repeating the style image $K$ number of times, where $K$ is the clip size for the input to the 3D CNN encoder. Both the content and style clips are first encoded with a 3D CNN encoder. The outputs of the encoder, i.e. final layer feature maps and the intermediate layer feature maps, are then fed to appearance subnetworks, which are responsible for extracting the appearance features only, disentangling them from the motion features. We will detail the method for achieving such disentanglement in the following sections. Once the outputs from appearance subnetworks are

available for both content and style clips, we perform the AdaIN3D (extension of AdaIN[11] operation to 3D CNN features, described in Section 3.2.3 below) operation for transferring feature statistics from the style clip features to the content clip features. Finally the intermediate layer features are directly fed to the decoder, which is also a 3D CNN. The final layer features after the AdaIN3D operation are then aggregated with the final layer feature maps from the 3D CNN encoder. This step is for re-inducing the motion features into the appearance features which were disentangled by the appearance subnetworks. These aggregated features are then fed into the Entangle Subnet, the output of this subnet is passed as input to the decoder which produces the desired stylized output clip. Effectively, these steps can be seen as, (i) disentangle the appearance features from the full features which contains appearance and motion both, (ii) stylize the appearance features only, and finally (iii) re-induce the motion component into the stylized appearance features and decode them to obtain the stylized clip.

## 3.2 Details of network components

Now, we will explain each component of the full architecture shown in the Figure 1. Please refer to the supplementary material for the complete code-level details of the appearance, Entangle, and the 3D Decoder subnetworks.

### 3.2.1 Normalized encoder

Our encoder is a state-of-the-art 3D CNN, i.e. C3D [25], which was originally proposed for the task of human action classification. It takes an input clip with a fixed number of $K$ frames, and then successively forward passes them through several 3D convolution and normalization layers. In the proposed network, this network is used twice with weight sharing, taking as input the content clip and the style clip respectively. The style image is repeated $K$ times to make a static video clip to input to the network, as using a 2D CNN to transfer feature statistics onto 3D CNN based content clip features for stylization did not give good results in initial experiments, as might be expected.

As we do feature statistics transfer between the content and the style feature vectors using AdaIN3D operation, we require the features of the layers of the C3D network to be of similar scale. This has been reported and solved before by doing network normalization by Gatys et al. [8]. Hence, we normalize the weights of our 3D encoder network using the UCF-101 dataset [22] following the method proposed by Gatys et al. [8].

### 3.2.2 Appearance subnets

In initial experiments, we attempted to do stylization by performing AdaIN3D operation between the features of content clip and style clip, as obtained from the C3D encoder. However, we noticed "slow motion" artifacts in the stylized video. We attribute those to the fact that motion and appearance are entangled in the C3D features, as observed and qualitatively visualized in the original C3D paper [25]. When we did stylization with a static style video (made by repeating style image), it led to inducing the static motion in the stylized clip. Hence, to do stylization with 3D CNN, we need to disentangle the appearance part from the features and stylize only those with the style image, and then recombine them with motion and decode them to the output stylized clip.

Hence, to extract the appearance component of the features from the C3D features, we use appearance subnetworks, which are themselves sequences of 3D convolution followed by ReLU layers. We use four appearance subnetworks which work with four different intermediate levels of the C3D decoder. We learn these networks in a different training phase, which we describe below when we detail the training procedure in Section 3.3.

### 3.2.3 AdaIN3D

The AdaIN operation proposed in AdaIN [11] was designed to work on features of 2D CNN. We propose a simple extension, for it to work with 3D features, as follows.

$$\text{AdaIN3D}(x, y) = \sigma(y) \left( \frac{x - \mu(x)}{\sigma(x)} \right) + \mu(y), \tag{1}$$

$$\mu_{nc}(x) = \frac{1}{THW} \sum_{t=1}^{T} \sum_{h=1}^{H} \sum_{w=1}^{W} x_{ncthw}, \tag{2}$$

$$\sigma_{nc}(x) = \sqrt{\frac{1}{THW} \sum_{t=1}^{T} \sum_{h=1}^{H} \sum_{w=1}^{W} (x_{ncthw} - \mu_{nc}(x))^2 + \epsilon}, \tag{3}$$

where $x$ represents the content clip and $y$ represents the style clip, and $n, c, t, h, w$ denote batch size, feature channels, temporal, height and width dimensions respectively. The mean and standard deviations of the style features in space and temporal dimensions are transferred to that of the content features.

### 3.2.4  Entangle subnetwork

In order to recover both motion and appearance details in the stylized video, we require an additional subnetwork to re-induce motion information to the appearance subnetwork features. To achieve this, we use an entanglement network which comprises of single pair of 3D Conv and ReLU layers. The input to the entangle layer is a weighted combination of the appearance subnet 4's output and the final output of the C3D encoder, i.e.,

$$x_e = \text{concat}(0.7x_a, 0.3x_m), \tag{4}$$

where, $x_a$ is the appearance feature map output of appearance subnet $4$ and $x_m$ is the feature map output of the `relu4_1` layer of the C3D network.

### 3.2.5  Decoder

The decoder is a 3D CNN used to generate the desired stylized video clip. It comprises of 3D Conv, ReLU, ReflectionPad3d and Upsample layers. The input of the decoder is the output of the Entangle Subnet, and the outputs of the Apperance Subnets are added to the decoder as skip connections by concatenation along the channel dimension of the decoder's appropriate intermediate feature maps.

## 3.3  Training

We propose a multi-phase training for learning the different subnetworks of the proposed network. In each phase, we train one or more subnetworks while keeping the other subnetworks fixed. We first detail the different losses used, and then explain the different phases of the training.

### 3.3.1  Losses used for training

We use different combinations of losses during different phases of training. In this section we give details of the losses, and specify which losses are used in which phases in the respective sections describing the training phases below.

The first loss we use is the standard $\ell_2$ reconstruction loss given by,

$$\mathcal{L}_{\text{reconstruction}} = \|O_r - I_c\|_2 \tag{5}$$

where $O_r$ and $I_c$ are the output reconstructed and input content clips respectively. This loss minimizes the $\ell_2$ distance between the input and the reconstructed content clip in the RGB pixel space.

Second loss we use is the $\ell_2$ loss on the output feature maps of the intermediate layers of the VGG-19 network, i.e.,

$$\mathcal{L}_{\text{appearance}} = \sum_i \|\phi_i(O_r) - \phi_i(I_c)\|_2 \tag{6}$$

where, $\phi_i$ is the feature map of layer $i$ of the ImageNet pretrained VGG-19 network. The intermediate layers of VGG-19 used, indexed by $i$, are `relu1_1`, `relu2_1`, `relu3_1` and `relu4_1`.

Most of the state-of-the-art style transfer methods [4, 11, 18] used features from the VGG-19 network pretrained on the ImageNet dataset. Since the VGG-19 features of individual frames only contain the appearance information, we use loss based on VGG-19 features to help disentangle appearance

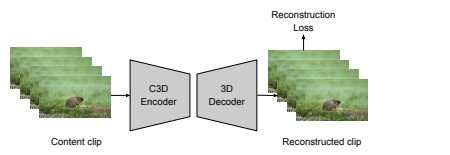

(a) First phase: Autoencoder training

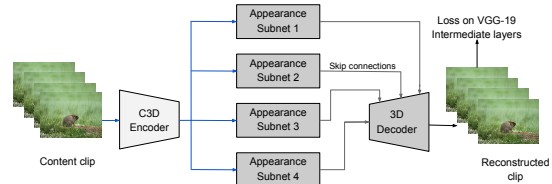

(b) Second phase: Appearance subnetworks training

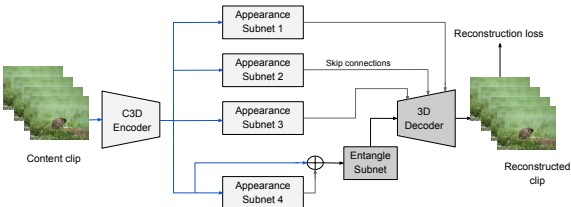

(c) Third phase: Entangle subnetwork training

Figure 2: The networks and settings for the different phases of training. The light gray subnetworks are fixed, while the different dark gray subnetworks are trained during the respective phases.

features from motion features in the C3D encoder. Further, we use content and style losses similar to AdaIN [11]. The content loss is given by

$$\mathcal{L}_{\text{content}} = \|\phi_i(O) - \phi_i(I_c)\|_2, \quad i = \texttt{relu4\_1}, \tag{7}$$

where $O$ and $I_c$ is the output stylized and input content clip respectively and the style loss is given by

$$\mathcal{L}_{\text{style}} = \sum_i \|\mu(\phi_i(O)) - \mu(\phi_i(I_s))\|_2 + \|\sigma(\phi_i(O)) - \sigma(\phi_i(I_s))\|_2, \tag{8}$$

where $i$ indexes over the $\texttt{relu1\_1}, \texttt{relu2\_1}, \texttt{relu3\_1}, \texttt{relu4\_1}$ layers of the VGG-19 network, $I_s$ is the input style clip, and $\mu(\cdot)$ and $\sigma(\cdot)$ are the mean and standard deviation of the feature maps.

We also use temporal loss, which is inspired by Gao et al. [6], and is given by

$$\mathcal{L}_{\text{temporal}} = \sum_{t=2}^{L} \sum_j \frac{1}{D_t} \|(O_t - W_t(O_{t-1}))_j - (I_{c,t} - W_t(I_{c,t-1}))_Y\|^2, \tag{9}$$

where $j \in \{R, G, B\}$ is each of the RGB channels of the image, and $Y$ is the relative luminance channel. $O_{t-1}$ and $O_t$ are the previous and the current stylized output frame, $I_{c,t-1}$ and $I_{c,t}$ are the previous and the current content input frame, $D_t = H \times W$ where $H, W$ are the height and width, respectively, of the input/output frames, and $W_t$ is the forward optical flow function, which is computed using FlowNet2.0 [12]. The purpose of using this loss is to maintain similar optical flow characteristics in each channel of the output frames, as that in the luminance channel of the input frame, and achieve motion consistency between frames.

Further, we propose an intra-clip loss given by,

$$\mathcal{L}_{\text{intra}} = \sum_i \sum_{t=2}^{L} \|\mu(\phi_i(O_t)) - \mu(\phi_i(O_{t-1}))\|_2 + \sum_i \sum_{t=2}^{L} \|\sigma(\phi_i(O_t) - \sigma(\phi_i(O_{t-1}))\|_2, \tag{10}$$

where, $i$ indexes over the layers of the VGG-19 network as above, $O$ is the stylized clip, $t$ represents the frame number in the stylized clip, $\mu(\cdot)$ and $\sigma(\cdot)$ are the mean and standard deviation of the feature map $\phi_i(\cdot)$. This loss encourages the feature statistics to be similar between successive frames of the stylized clip. By ensuring that, we mitigate the effect of the frames being stylized differently, even when the style image is the same, and avoid jerky changes in color and intensity.

### 3.3.2 First phase: Autoencoder training

The first phase of training is to train the C3D encoder and decoder for video reconstruction task. The original C3D network was proposed for a discrete classification task, while the task here is of dense

prediction of stylized output video. To have a suitable initialization for the encoder and decoder, we pretrain them with reconstruction loss, Eq. 5, on our training dataset of videos, with the network shown in Figure 2a.

### 3.3.3 Second phase: Appearance subnetworks training

In the second phase, we keep parameters of all appearance subnetworks as well as the decoder trainable, while keeping the parameters of the encoder fixed. The network architecture is shown in Figure 2b. We pass the features from different intermediate layers of the encoder to the corresponding appearance subnetworks for extracting out the multi-scale appearance information from the entangled feature maps of the encoder. The output of the appearance subnetwork 4 is passed as the input to the decoder while outputs of other appearance subnetworks are passed as skip connections, concatenated channel-wise, to the decoder. We initialize the decoder with the weights learned from the first phase for the second phase. We use ImageNet pretrained VGG-19 intermediate feature maps reconstruction loss, Eq. 6, during this phase. This ensures that the appearance subnetworks together with the decoder capture the appearance part of the input video clip, by reconstructing the VGG-19 feature maps. Since this loss is applied independently for each frame of the input clip, it mainly captures the appearance information and discounts the motion information, leading to disentanglement of the appearance and motion features by the appearance subnetworks.

The disentanglement of motion and appearance, by the four appearance subnets, happens here in the phase 2 training. In this phase, we keep the 3D CNN encoder fixed, so the features which are input to the appearance subnets have entangled motion and appearance information. We train the network to minimize the VGG-19 feature loss for each frame, and hence only remember appearance information in the frames. When trained like that, the appearance subnets only gate the appearance information from the 3D encoder features as only that is required to minimize the loss, which is applied independently to each frame and, thus, has no dependence on motion. The evidence for such disentanglement is indirect, as when we do not do this, we get motion artifacts which we explain by the static nature of the style clip (constructed by repeating the style image), but when we do this, those motion artifacts disappear indicating that the features and the AdaIN3D based statistics transfer, only affected the appearance and not the motion.

### 3.3.4 Third phase: Entangle subnetwork training

In the third phase, we keep the decoder and the entangle subnetwork parameters trainable, while keeping the parameters of all appearance subnetworks as well as the encoder fixed. We initialize the decoder from scratch for this phase as the input features are different from the previously trained decoders. We use reconstruction loss for this phase. The network architecture is shown in Figure 2c. The aim of this phase of training is to learn to re-induce motion information into the appearance features.

### 3.3.5 Fourth and final phase: 3D decoder training

In the final phase, the full network is used, and only the decoder parameters are trained. We use the decoder trained in the third phase as the initialization for the decoder in this phase. All the other modules, i.e. encoder, all four appearance subnetworks, and entangle subnetwork, are kept fixed. The aim is to learn a final decoder which is able to do stylization. The combination of losses gives the full loss used for this phase, Eq. 7, 8, 9, 10,

$$\mathcal{L} = \sum_i \lambda_i \mathcal{L}_i, \text{ where, } i \in \{\text{content, style, temporal, intra}\}, \tag{11}$$

i.e. this phase learns to do stylization while keeping intra-clip consistency, preserving overall content, and transferring style. This phase completes our training.

## 4    Experiments

We now describe the experimental setup, and the qualitative and quantitative results comparing the proposed method with existing state-of-the-art approaches.

### 4.1 Implementation Details

**Dataset.** A video dataset was not available in the existing style transfer literature for video stylization. Hence, we propose a new dataset curated from existing public benchmark datasets, to train for video stylization task. We will make the links and all details of the dataset public for use by the community.

Since, most of the challenge in video stylization stems from motion in the videos, we explicitly construct the dataset such that the clips have a high amount of motion, and are not just static repeated images. On analysing some videos downloaded we saw that some videos were simply repetition of a single static image, and were thus not appropriate.

We downloaded $45,574$ videos from the first $100,000$ URLs in the Sports1M dataset [14] (as many links were now defunct, and some downloads failed). We then generated 16 frame clips with a high amount of motion with the following steps. First, we randomly sampled a video from the 45K videos dataset, and used deep learning based shot detection by Souček et al. [23] to split the video into shots. We then randomly sample a 16 frame clip and computed the Farneback optical flow [5] of the frames of the sampled clip. We then selected the clips which have an average optical flow value above a threshold to be in the proposed dataset. Overall, we used $10,000$ content clips generated from the above procedure as part of the dataset for training. In addition to the content clips, we use images from the WikiArt [20] as the style images, as has been done by the image stylization methods as well.

**Input.** We set clip size $K$ to 16. While training we extract random $128 \times 128$ size same patch from consecutive frames of a video clip in case of content clip. For, style clip we extract a random $128 \times 128$ patch from the style image and repeat it 16 times which is equal to the length of the content clip. During testing, we pass the video in chunks of successive 16 frames clips.

**Parameters.** In the first phase of training, we train the encoder and decoder for 10k iterations. We used Adam optimizer with a learning rate of $10^{-4}$ with a decay rate of $5 \times 10^{-5}$ after every iteration for this phase of training. In the second phase of training, we train the decoder, appearance subnets for 5k iterations. We used Adam optimizer with a learning rate of $10^{-4}$ with a decay rate of $5 \times 10^{-5}$ after every iteration for this phase of training. In the third phase of training, we train the decoder and entangle subnet for 100k iterations. We used Adam optimizer with a learning rate of $10^{-4}$ with a decay rate of $5 \times 10^{-5}$ after every iteration for this phase of training. In the final phase training, we train the decoder for 40k iterations with content, style loss where values of $\lambda_c, \lambda_s$ is $1, 2$ respectively. After this, we train the decoder with content, style, temporal losses for 160k iterations where values of $\lambda_{\text{content}}, \lambda_{\text{style}}, \lambda_{\text{temporal}}$ being $1, 2, 10$ respectively. And finally, we fine-tune the decoder for 40k iterations with content, style, temporal, intra-clip loss where values of $\lambda_{\text{content}}, \lambda_{\text{style}}, \lambda_{temporal}, \lambda_{\text{intra}}$ being $1, 2, 10, 10$ respectively.

### 4.2 Ablation Studies

We discuss the takeaways from the ablation experiments we performed. It is difficult to show the video related effects with images. Kindly, see the video results in the supplementary materials corresponding to these discussions.

**Disentangling appearance and motion using appearance subnetworks.** We applied AdaIN3D operation directly to the encoded tensor of the content and style clips by the C3D network without using any appearance (disentanglement) subnetworks. However, we observed jerky motion in the generated videos, which followed a repeated pattern across the clips. Upon using appearance subnetworks, such motion artifacts were not observed anymore. The supporting videos are in supplementary material folder `Videos_with_naive_extension_of_2D_stylization`.

**Temporal loss.** When trained without temporal loss, the stylized video has two kinds of artifacts. First, color consistency across subsequent frames suffers as we transfer style with appearance subnetworks which remove motion component. Second, there is some loss of motion as well. Both of these artifacts are not observed, or are very mild once we use temporal loss. Supporting videos are in supplementary material folder `Videos_without_temporal`.

**Intra-clip loss.** We observe even after applying the temporal loss, there are some color inconsistencies leading to flashing like artifacts. However, when we use intra-clip loss such inconsistencies are addressed to a large extent. Supporting videos are in supplementary material folder `Videos_without_intra`.

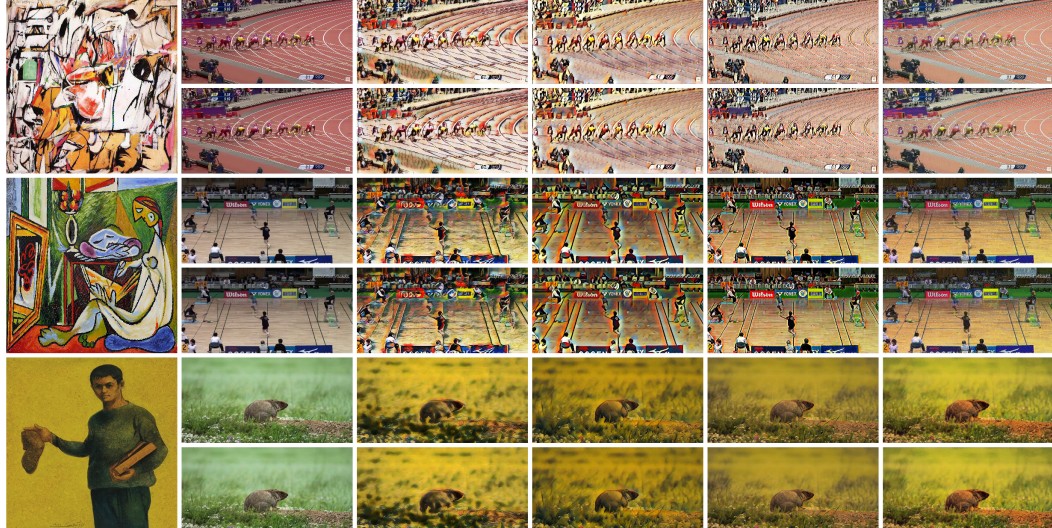

Figure 3: First column is the style image and subsequent columns are a pair of frames generated using different approaches, in the following order: AdaIN[11], MCCNet[3], AdaAttn[18] and Ours.

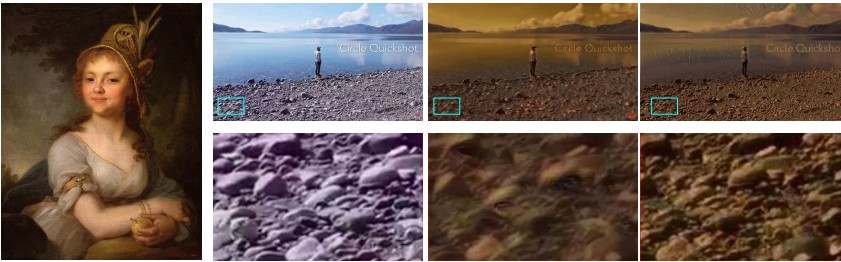

Figure 4: Effect on content of different methods. First column is the style image, and the rows in the remaining columns contain frames and zoom-ins of original content clip, AdaAttN [18] stylized clip and our method. Notice how AdaAttN (middle) modifies the content significantly, while the proposed method (right) tends to keep the content relatively preserved.

## 4.3 Qualitative results

Figure 3 shows some qualitative results of the proposed method compared to existing methods. We observe that the proposed method focuses more on low level texture of the style image while AdaAttn [18] tends to flatten out the textured regions consistently. Figure 4 shows that AdaAttn also tends to modify the content, while the proposed method gives better textured outputs closer to the original content. Please see supplementary material for video results. Overall the results produced by the proposed method are aesthetically pleasing, and can be used by video creatives as one of the filters available.

### 4.3.1 User Study

We considered AdaIN [11], MAST [4] and two state-of-the-art methods, MCCNet [3] and AdaAttN [18], for comparison with our method. We randomly picked 15 content clips and 15 style images and considered random 20 stylized videos out of 225 possible combinations. We generated these stylized videos using all the compared methods. We then presented the stylized video of other methods along with that of our method side by side, but in random order, to the participants. We collected 900 votes from

| Method | Content | Style | Overall |
|--------|---------|-------|---------|
| AdaIN | 2.3 | 16.3 | 4.3 |
| MAST | 1.0 | 15.7 | 11.0 |
| MCCNet | 12.3 | **31.3** | 27.7 |
| AdaAttN | 41.0 | 20.0 | 28.3 |
| Ours | **43.3** | 16.7 | **28.7** |

Table 1: User study preferences(%). **Bold**, red, blue represent rank 1, 2 and 3 respectively.

| Method | 1 | 2 | 3 | 4 | 5 | 6 | 7 | 8 | 9 | 10 | 11 | 12 | 13 | 14 | 15 | 16 | 17 | 18 | 19 | 20 | Mean |
|---|---|---|---|---|---|---|---|---|---|---|---|---|---|---|---|---|---|---|---|---|---|
| AdaIN | 0.95 | 3.46 | 0.56 | 4.88 | 0.64 | 1.06 | 0.66 | 3.4 | 3.68 | 1.72 | 1.77 | 6.22 | 0.32 | 2.85 | 0.13 | 0.34 | 1.31 | 5.64 | 0.85 | 0.45 | 2.04 |
| MAST | 1.89 | 3.59 | 1.6 | 4.77 | 1.17 | 1.53 | 0.79 | 4.14 | 3.6 | 2.17 | 1.56 | 6.28 | 1.06 | 3.45 | 0.44 | 0.74 | 0.97 | 5.08 | 1.11 | 0.89 | 2.34 |
| MCCNet | 0.59 | 2.21 | **0.3** | 3.06 | 0.43 | 0.85 | 0.44 | 2.29 | 2.45 | 1.07 | 1.51 | 5.8 | 0.21 | 3.16 | 0.09 | 0.15 | 0.91 | 4.44 | 0.57 | 0.21 | 1.53 |
| SANET | 0.88 | 3.01 | 0.95 | 4.21 | 0.78 | 1.45 | 0.62 | 3.94 | 3.16 | 1.99 | 1.6 | 4.42 | 0.5 | 2.96 | 0.23 | 0.63 | 1.11 | 4.73 | 0.89 | 0.55 | 1.93 |
| AdaAttN | 0.39 | 1.96 | 0.39 | 2.94 | 0.39 | 0.64 | 0.29 | 2.25 | 1.77 | 0.98 | 0.83 | 5.51 | 0.19 | 2.91 | 0.07 | 0.08 | 0.34 | 2.69 | 0.36 | 0.24 | 1.26 |
| Ours | 0.65 | 3.98 | 1.36 | 4.57 | 0.32 | 0.65 | 0.85 | 2.79 | 2.92 | 1.64 | 0.76 | 3.58 | 0.41 | 2.01 | 0.15 | 0.2 | 0.69 | 4.29 | 0.55 | 0.35 | 1.63 |
| Ours+temporal | **0.37** | **1.89** | 0.31 | **2.45** | 0.27 | **0.43** | 0.44 | 2.17 | 1.97 | **0.94** | 0.59 | 1.96 | 0.22 | **1.58** | 0.08 | 0.15 | 0.42 | 3.25 | 0.41 | **0.21** | **1.00** |
| Ours+temporal+intra | 0.41 | 2.18 | 0.31 | 2.59 | 0.33 | 0.5 | 0.44 | **1.7** | 2.6 | 1.06 | 0.66 | **1.89** | 0.21 | 1.92 | **0.07** | 0.13 | 0.53 | 3.93 | 0.51 | 0.28 | 1.11 |

Table 2: Optical flow evaluation 20 test pairs of content videos and test images. **Bold**, red and blue color represents the rank 1, 2 and 3 minimum error respectively.

various participants for content preservation, degree of stylization and overall preference. The results in Table 1 show that our method leads for content preservation closely followed by AdaAttN [18]. In the case of the degree of stylization, MCCNet [3] leads us by a margin, but we argue that there is always a trade off between style transfer and content preservation, and so we can observe that MCCNet [3] performs poorly in the case of content preservation. Considering the overall preference in which participants have considered both content and style, our method performs better than all the other methods, proving our method's effectiveness as a successful alternate style transfer method.

### 4.4 Quantitative Results

We computed Farneback optical flow [5] for the content video and the stylized video. We report the mean of the absolute difference between the flows of content video to that of corresponding flow in stylized video. This is an approximate measure to ensure that the stylization method is preserving the motion in the original clip. All the methods are expected to distort the motion in some ways while performing stylization, and the degree of such distortion will depend on the input video. We observe that the proposed method achieves competitive flow errors.

### 4.5 Compute time

Since the proposed method is based on 3D CNNs, it is computationally heavier than the 2D CNN methods. To get a real world computation time estimate, we processed a video with 144 frames with resolutions of 640x360 pixels. AdaAttN took 14 seconds and 8GB of GPU memory, while our proposed method took 60 seconds and 16GB of GPU memory on a machine with Intel Core i9-10900X processor and Nvidia RTX A4000 GPU.

## 5 Conclusion

We proposed a novel architecture and training method for the task of video stylization given a reference style image. The method uses 3D CNNs exclusively, and is the first to report stylization with them. It achieves stylization by disentangling appearance information from the 3D CNN based encoder's features, stylizing them, and then re-combining them with the motion information before finally decoding to the desired stylized output video. The training is done in phases with different phases ensuring different capabilities for the subnetworks in the architecture. We also proposed a video dataset containing videos with high degree of motion for training video stylization networks. We showed qualitatively and quantitatively that the proposed method produces results which are competitive to the existing methods and produce different and aesthetically pleasing results. We hope the proposed method will result in another filter in video creative's toolbox.

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
