# OpenReview forum: "ViSt3D: Video Stylization with 3D CNN"
_NeurIPS.cc/2023/Conference — NeurIPS 2023 poster_

### Official Review · Reviewer_qZ5E · 2023-06-26

**Soundness:** 3 good
**Presentation:** 3 good
**Contribution:** 2 fair
**Rating:** 4
**Confidence:** 4

**Summary:**

This paper proposes a 3D CNN based video stylization method which explicitly disentangles motion and appearance and adopts multi-phrase training. Experiments show that the method achieves high quality results.

**Strengths:**

The proposed 3D CNN based framework and multi-phrase training is novel and effective, the paper also expand the AdaIN in time dimension to AdaIN3D. The quality of stylization results is also good.

**Weaknesses:**

I think the main weakness lies in the experiments.
1. In ablation study, the authors only showed and described the results but didn't not analyze the reason
2. In the qualitative results, this paper compares with only a few previous methods, while some related works should also be discussed. for example,
Kotovenko D, Sanakoyeu A, Lang S, et al. Content and style disentanglement for artistic style transfer[C]//Proceedings of the IEEE/CVF international conference on computer vision. 2019: 4422-4431.
Kotovenko D, Sanakoyeu A, Ma P, et al. A content transformation block for image style transfer[C]//Proceedings of the IEEE/CVF Conference on Computer Vision and Pattern Recognition. 2019: 10032-10041.
Both papers showed high-quality video stylization results
3. For tasks where the quality is largely depended on subjective assessment, user study is a common criterion. For this paper, I feel the comparison is not sufficient and a user study is needed
4. In quantitative comparison, the proposed method doesn‘t obviously outperform previous method.
5. Lack a comparison to single image stylization with optical flow motion compensation



**Questions:**

1. Why are only three previous methods included in the qualitative and quantitative comparison? There have been quantities of style transfer papers every year and many of them are about video stylization.
2. Why is user-study not included in the experiments?
3. What is the advantage compared to adding optical flow motion compensation to single frame style transfer method?


**Limitations:**

This paper lacks several important parts:
1. The discussion and analysis of each part of the method are not included in the paper
2. User study is missing in experiments
3. More previous methods should be included in comparison
4. It would be better if more results of diverse scenarios could be shown.

---

> ### Author Rebuttal · Authors · 2023-08-10
>
> Thank you for the valuable comments, we answer your questions and concerns as follows.
>
> ### Why only three methods compared
>
> Video stylization is relatively less popular cf. image stylization. We reported the recent methods which are leading in this task to the best of our knowledge. We have further included quantitative results from two more papers in the comparisons table (final table provided in global comments), but as expected they perform below the leading works we had already cited and compared with. In the absence of specific citations, we are unable to give any better answer.
>
> ### Ablation study, results not analyzed
>
> The ablation studies basically support the motivations given in our approach section. Due to lack of space we could not reiterate them, we will do so in camera ready. We will also point out specifically that there are supporting videos in the supplementary material (showing before after results) originally uploaded with the paper.
>
> To summarize again:
>
> - First ablation shows that when the appearance and motion components in the features are not disentangled the results have jerky motion artifacts as the static style clip induces zero movement in the content clip. Once we do the disentanglement this is resolved. The supporting
> videos are in supplementary material folder Videos_with_naive_extension_of_2D_stylization
>
> - Second ablation shows that temporal loss is needed to make sure that in videos pixel motion consistency is maintained. Without the loss we get jittery movements in videos, but adding the temporal loss resolves this. Supporting videos are in supplementary material folder Videos_without_temporal.
>
> - Third ablation shows that without intra clip loss, there is a slight drift in colors as the stylization is not sufficiently tied for the different frames of the clip. But with intra clip loss that is resolved. Supporting videos are in supplementary material folder Videos_without_intra.
>
> ### User study
>
> Not all previous works have reported a user study. Based on your kind suggestions we did a user study during the rebuttal period and are reporting it in the global response which we request you to kindly refer to. In summary, the proposed method is competitive wrt existing methods in terms of user preferences as well, while being a first demonstration of stylization using 3D CNNs.
>
> ### Comparison with methods
>
> In the short period of time we could not do a comparison to the methods specifically mentioned in the review. Their code or pre trained models for inference were not available.
>
> However, we were able to compare with SANet and MAST quantitatively, and also used these in the user study (so compared qualitatively).
>
> We iterate again however, the task of stylization is not a well defined task with a single true output for a specific (content, style) pair. We have demonstrated that the proposed method is competitive to the existing methods in terms of quantitative metrics, as well as subjective qualitative results and a user study.
>
> ### Advantage cf. adding optical flow to image stylization methods
>
> We also found another paper which does specifically this [A], i.e. adds optical flow to 2D CNN based image stylization methods. We have added a video with comparison to some of their results. We observe that in the first example, there is a large amount of flickering in the first example generated with [A], while in the second example the method smoothes out the texture of the pebbles on the beach, while the proposed method does not have heavy flickering, and it also maintains the texture of the pebbles.
>
> The two approaches give different results which may be used by practitioners in different use cases.
>
> [A] Wang et al., Consistent Video Style Transfer via Relaxation and Regularization, Trans. Image Processing, 2020
>
>
> ### More previous methods in comparison
>
> We had compared with AdaAttn which had itself compared to many methods so we hoped demonstrating better results wrt AdaAttn would be sufficient. However, at your suggestion we have included two more methods in the comparison table, and have given the updated table in the global response. Kindly refer to that please.
>
>
> ### More results in diverse scenarios
>
> Due to space limitation we couldn’t include more visual results in the main paper PDF.  But, we had already included a very variety of results in the supplementary which we summarize here: in the folder "Comparative Analysis” we had given 4 video, each containing 3 (input video, target style image) pairs for our method as well as AdaIn, MCCNet and AdaAttn. These covered indoors, outdoors, high motion, animation and sports. In addition, in the folder ``Our results” we provided 6 result videos of a one input video each (4 short videos and 2 longer videos) stylized with three different style images, these also cover a variety of scenes and styles. We request you to kindly refer to the supplementary results videos. In addition we used 20 videos for the user study with 1 style each, which the users found reasonable as well (user study reported in global author response).

---

### Official Review · Reviewer_fJLx · 2023-07-04

**Soundness:** 2 fair
**Presentation:** 2 fair
**Contribution:** 2 fair
**Rating:** 5
**Confidence:** 3

**Summary:**

Image stylization is more popular than video stylization, research on video stylization is few due to its challenging. This manuscript first applied 3DCNN to video stylization task, it first explicitly disentangles motion and appearance, and then stylizes the appearance part, and then adds back the motion to decode the final stylized video. The method is trained in multiple phases. Experimental results show the superiority of the proposed method with existing methods. Hwever, there are still some problems in this manuscript.

**Strengths:**

Stylization has been a very popular research area, however video stylization is more challenging than image stylization. Most existing methods directly apply image stylization methods to videos by processing the videos frame by frame, results are not well. This manuscript proposes a novel method based on 3DCNN and achieves better results than existing methods. Moreover, a large-scale dataset with 10, 000 content clips curated from public benchmark Sports1M is built for this task.

**Weaknesses:**

The task is interesting and the proposed method has some improvement, but paper writing is a little weak. The logic of the article is confusing, especially in Section 3 APPROACH. The description and details of network structure is not clear, it's not easy to understand.

**Questions:**

1.	How to disentangle motion and appearance by 4 appearance subnets is not clear in your description.
2.	In line 190, ‘O and I are the output and input clips respectively’, and in line 213, ‘O is the stylized clip’. It is confusing whether they are content or style clip?
3.	Combined with the previous question, in Fig.2, it can be found that in all three phases, only content clip is used, where is the style clip? How to add style information to appearance and preserve motion?
4.	In line 134, ‘Our encoder is a recent state-of-the-art 3D CNN, i.e. C3D [23]’, while C3D was proposed in 2015, ‘recent’ may be not suitable.
5.	There are some fonts are inconsistency, e.g., line 172 ‘Appearance Subnet 4’, line 202 ‘relu1_1, relu2_1, relu3_1, relu4_1”.


**Limitations:**

1.	The authors point out that some flashing still occurs in challenging edge cases.
2.	In Table 1, it can be found that the proposed method did not achieve minimum error on all videos, which means it cannot preserve the motion in the original clip very well.
3.	From the video results in supplementary material, it can be seen that the degree of stylization by the proposed method is not obvious, and the results still have some artifacts.
4.	It is suggested to reorganize section 3 to introduce your method more clearly.

---

> ### Author Rebuttal · Authors · 2023-08-10
>
> Thank you for the valuable comments, we answer your questions and concerns as follows.
>
> ### Disentangle motion and appearance by 4 appearance subnets
>
> The disentanglement of motion and appearance, by the 4 appearance subnets, happens by the phase 2 training. In Phase two we keep the 3D CNN encoder fixed, so the features which are input to the appearance subnets have entangled motion and appearance information. We train the network to minimize the VGG-19 feature loss for each frame, and hence only to remember appearance information in the frames. When trained like that the appearance subnets only gate the appearance information from the 3D encoder features as only that is required to minimize the loss, which is applied independently to each frame and has no dependence on motion. The evidence for such disentanglement is indirect, as when we do not do this, we get motion artifacts which we explain by the static nature of the style clip (constructed by repeating the style image), but when we do this those motion artifacts disappear indicating that that the features and the AdaIN3D based statistics transfer only affected the appearance and not the motion.
>
>
>
> ### In line 190, ‘O and I are the output and input clips respectively’, and in line 213, ‘O is the stylized clip’. It is confusing whether they are content or style clips?
>
> We apologize for the confusion.
>
> The losses as functions of two variables, $I$ and $O$ are correct and what those variables are depends on the training phase.
>
> To clarify, in L190 Eq.5 the reconstruction loss applies to phase 1 training of the autoencoder, so the two inputs to the loss function are the input clip, and the output reconstructed clip.
>
> While in L213, Eq.10 the intra clip loss applies to phase 4 of the training of full stylization, so the inputs to the loss function are the frames of the output stylized clip.
>
> Nonetheless, to avoid confusion, we will change $O$ in L190 to $O_r$, as it is the reconstructed output clip from the auto-encoder.
>
>
> ### How is style clip used and how is appearance and motion preserved while doing stylization?
>
> Style clip is only used in the final phase of training shown in Fig.1. Style information is added using AdaIN3D operation, which generates a combined feature by taking content, and style features as input. The AdaIN3D operation introduces style to disentangled appearance features only, which are then combined with motion features. The combined features are then decoded and the final losses (content, style, temporal and intra-clip, Eq11) make sure that the output is close in appearance to the content clip, while having style of the style clip, and at the same time not having jittering and flashing artifacts.
>
>
> ### C3D proposed in 2015, ‘recent’ may be not suitable.
>
> We will remove recent from the text mentioned.
>
> ### Font inconsistency, e.g., line 172 ‘Appearance Subnet 4’, line 202 ‘relu1_1, relu2_1, relu3_1, relu4_1”.
>
> We wanted to highlight that those words indicate certain important layers’ features in the VGG-19 networks that are used. We will clarify this in the notations section.

---

> > ### Comment · Reviewer_fJLx · 2023-08-15
> >
> > Some questions have been solved and additional results are provided, I will keep my rating.

---

> > > ### Author Response · Authors · 2023-08-16
> > >
> > > Thank you again for your time and feedback on the paper, and reading the reviews.

---

### Official Review · Reviewer_9xQH · 2023-07-05

**Soundness:** 3 good
**Presentation:** 3 good
**Contribution:** 3 good
**Rating:** 6
**Confidence:** 3

**Summary:**

The paper studies the task of video stylization. The paper aims to stylize the video using 3D CNN and AdaIn3D. To perform the stylization, the authors propose to disentangle motion and appearance first, and then stylize the appearance part using AdaIn 3D. The results show state-of-the-art results compared to the baseline methods.

**Strengths:**

1) The paper proposes a novel framework using 3D CNN to perform the stylization. To achieve this, multiple designs are proposed. Such as the appearance subnet, and Entangle network.

2) It is reasonable to train the model in different stages to enforce the corresponding networks to learn their functionalities.

3) The results are significantly better compared to the baseline methods.

4) It is non-trivial to train a model with many sophisticated designs and achieve plausible results.

**Weaknesses:**

1) It is unclear why four appearance subnets are needed. If the number of appearance subnets is cut down, how the performance is affected?

2) Why the Entangle Subnet is necessary? What if the feature maps fed into the decoder are the weighted average of the output of the appearance network and C3D encoder?

**Questions:**

Please see my questions regarding the details of the network in Weakness.

**Limitations:**

The authors have discussed the limitations.

---

> ### Author Rebuttal · Authors · 2023-08-10
>
> Thank you for the valuable comments, we answer your questions and concerns as follows.
>
> ### Unclear why four appearance subnets needed, affect on performance if the number of appearance subnets is reduced
>
> The different appearance subnets work at different scales. The C3D architecture can be seen as four sequential logical blocks, and we used one appearance subnet for the output features of each of those blocks. This was inspired by skip connections architecture in many pixel to pixel networks, e.g. UNet for segmentation etc.
>
> If we remove the appearance subnets, we would observe a deterioration in the stylization quality along with undesirable artifacts. In C3D features qualitatively it has been shown that appearance contributes more to the lower features (initial blocks) while motion contributes more to the higher features. If we remove the lower layer appearance subnet we would observe distorted appearance in the output while if we remove the upper layer, motion artifacts will corrupt the output.
>
> ### Why is Entangle subnet needed, decoder features could be weighted average of appearance net and C3D encoder features
>
> Other than needing a simple projection layer to match different feature dimensions, a weighted average as you suggest could be a valid candidate for feature fusion. The Entangle network we use is very lightweight, i.e. just one 3D conv and one ReLU layer, and it effectively does a non-linear combination cf. a linear weighted average, ensuring the fusion is more effective.

---

> > ### Author Response · Authors · 2023-08-20
> >
> > Thank you again for your comments and feedback, we are happy to answer any more questions you might have. Thank you for your time.

---

> > ### Comment · Reviewer_9xQH · 2023-08-20
> >
> > I have read the rebuttal provided by the authors. The rebuttal addresses my questions regarding the details. I keep my rating as Weak Accept.

---

### Official Review · Reviewer_Syan · 2023-07-07

**Soundness:** 2 fair
**Presentation:** 3 good
**Contribution:** 2 fair
**Rating:** 5
**Confidence:** 4

**Summary:**

This paper proposes ViSt3D, which utilizes 3D CNN (C3D) as the encoder backbone for video style transfer. However, the motion and appearance information in C3D is intrinsically entangled. To address this problem, ViSt3D aims to separate these two features with appearance subnets and AdaIN3d. Results on sports1M datasets are used to demonstrate the performance of the proposed method.

**Strengths:**

- As claimed, this paper is the first to use 3D CNN for video stylization.

- The temporal and intra-loss improve the stylization stability as shown in the supplementary videos.

-  Quantitative evaluations of optical flow demonstrate the effectiveness of the proposed method.

**Weaknesses:**

- The usage of 3D CNN is not well motivated. The authors should put more emphasis on the motivations and advantages of using 3D CNN for stylization. This also makes the technical contributions a bit weak. And as mentioned in the limitations 3D CNN also brings extra computational cost compared with 2D CNN-based methods.

- In the paper, the style clip is formed by repeating a static image multiple times. Thus, I do not understand why you need a C3D to extract the style feature -- a 2D CNN could be sufficient. I suggest the authors conduct related experiments and provide thorough discussions.

- In Equation (10), why not use the difference of warped features to measure the intra-clip consistency as (9)? I think using the mean and standard deviation will hurt the results when the scene changes or there is a large motion.

- The model needs to be trained with four cascaded phases. This makes the model less elegant and may be hard to train and reproduce.

- Comparison with SOTAs should be more comprehensive. Only the optical flow metric is not enough to evaluate the style transfer results (for example, directly outputting the content video will result in a relatively small error).

**Questions:**

The authors are encouraged to address the weaknesses mentioned above, especially the motivations and contributions for using 3D CNN. If a 2D CNN is enough to address the stylization, it would be unnecessary to further design a disentangled module to solve a problem that is originally not there.

**Limitations:**

The authors discussed the limitations of the increase in both inference time and memory consumption. Although it may not be appropriate to directly compare the time and memory requirements of our approach with other methods based on 2D CNNs, it would be still helpful to give related statistics for a better understanding of the proposed method.

---

> ### Author Rebuttal · Authors · 2023-08-10
>
> Thank you for the valuable comments, we answer your questions and concerns as follows.
>
>
> ### 3D CNN vs 2D CNN, solved problem, motivation
>
> Stylization is not a deterministic problem, in the sense that there is no single correct stylization for a (input content, target style) pair. It is akin to a high level filter which transfers characteristics from an image to another image or video. 3D CNN was a natural choice when working with videos. They had never been used for the task, and we expected to obtain different stylization characteristics upon using them, as eventually we did and reported in the paper. 2D CNNs have also been used with some success for the task, but none of the methods (2D or 3D CNN based) could be THE final solution. 3D CNNs do take more computational resources but they produce competitive and useful results as we reported in the paper and reinforce in the user study provided in the other response. Hence we believe our contribution of successfully doing stylization with 3D CNNs for the first time, is valuable for the research community as well as practitioners. The simple motivation of using 3D CNNs was to determine if stylization can be done with 3D CNN at all (which surely in hindsight, given the paper, looks obvious now, but was a hard problem to solve) and to derive a novel and different stylization which is useful.
>
> ### Why 3D CNN for style clip and not 2D CNN
> Stylization requires transfer of feature statistics from style image to the content video/image. Just as for image stylization using one model for encoding the content image (e.g. ResNet) and another for encoding the style image (e.g. InceptionNet) is not expected to lead to sensible results, as the statistics for intermediate features are expected to be very different to be successfully transferred, in the present case using two different networks for encoding content and style assets is not expected to work as well. We did initial experiments with 2D CNN encoding for style images, as a sanity check, but as expected it failed completely, i.e. the outputs did not converge to reasonable videos. Hence we moved on to encoding the style image with 3D CNN, by making a static clip, which was the naive extension to AdaIN with 3D CNNs for content and style both. We provided those results in the supplementary materials folder Videos_with_naive_extension_of_2D_stylization.
>
> Please note similar repetition (as used for single style image), albeit for features, was used to bootstrap/initialize the 3D CNN parameters in the one of the 3D CNN for video classification papers, and so is not a new concept and has been used successfully in the past.
>
> ### Intra-clip loss, temporal loss, high motion, scene change
> The intra-clip loss in Eq.10 is to match the global color appearance of successive frames. While the temporal loss in Eq.9 is to preserve pixel level motion characteristics after considering the appropriate warping of frames. Even in case of large motion, since Eq.10 works with global averaged values, it successfully performs its job. We had provided qualitative results without intra-clip loss (folder Videos_without_intra) and also without temporal loss (folder Videos_without_temporal), in the supplementary material. Without temporal loss, we observe strong jittering and without intra clip loss we observe a subtle flashing, as the average color of the frames drifts, and then resets on a clip boundary. Adding the intra clip loss fixes this flashing.
>
> Large sudden motions are challenging for all stylization methods, we had provided qualitative comparisons in supplementary (folder Comparative Analysis, result 2 has a large motion clip from Spiderman movie, and result 4 has a clip from Avengers movie). The proposed method does competitively or better than existing methods.
>
> None of the video stylization methods address shot changes where the scene completely changes. The stylization is done on videos which have been separated by performing shot detection first as we explain in our dataset creation section (l265).
>
>
> ### Model/training with four phases, less elegant, not reproducible
> We agree that the current version of the model and training are quite involved. The current paper demonstrates that stylization can be done with 3D CNNs. In the work we are currently doing, we are shrinking the models in size and experimenting with end-to-end training with promising initial results. As a first result, we believe the current method is interesting for the community. We had already given the code of each subnetwork in the supplementary PDF, and each stage of training is a well understood training procedure for the community (auto-encoder, perceptual and reconstruction loss minimization), so we believe that the results can be reproduced by a reasonably skilled student researcher.
>
> ### Comparison to SOTA, optical flow not good/sufficient metric
> We agree to this point, but we highlight that evaluating a subjective task is always hard, and some proxy metrics are generally used. Stylization, like some other tasks, perhaps (semantic) edge detection, image captioning etc., do not have one true answer. So using proxy metrics to evaluate aspects of the output have been used by the community which we also follow. In addition, as another reviewer’s question’s response, we also did a user study in the present rebuttal, and we request you to refer to that as well. In particular among responses rating for preservation of content and transfer of style, we see a trade-off. The methods which preserve the content (in extreme case output the same video as you mention) lag in transferring style. Overall the proposed method strikes a balance as good as or better than the existing methods, while being a completely novel way of doing stylization.
>
> ### Time and memory cf. 2D CNN
> We processed 144 frames of 640x360. AdaAttN took 14, 8GB of GPU memory, while proposed method took 60s, 16GB of GPU memory on a machine w/ Core i9-10900X & A4000 GPU.

---

> > ### Author Response · Authors · 2023-08-16
> >
> > Thank you again for your time and feedback. Kindly let us know, if you have any further questions after the rebuttal.

---

> > > ### Comment · Reviewer_Syan · 2023-08-20
> > >
> > > I appreciate the detailed rebuttal provided by the authors, which resolved several of my concerns (using 3D CNN for style, user study, and reproductivity). Addressing the issues raised by other reviewers also convinced me of the paper's significance. Thus, I improve my initial rating. I am still a bit concerned about the technical contribution of this paper -- the key to the success of using 3D CNN seems to be a complicated designed training pipeline.

---

> > > > ### Author Response · Authors · 2023-08-20
> > > >
> > > > Thank you very much again for taking the time to review the paper, and then going through the rebuttal. We will incorporate your feedback in the camera ready paper, and supplementary material.

---

### Author Rebuttal · Authors · 2023-08-10

We thank the reviewers and ACs for their valuable time and constructive comments on the paper.

The reviewers have raised many valid points and concerns which we have answered to the best of our abilities and hope that the reviewers will find them satisfactory.

We would like to reiterate that video stylization with 3D CNNs was not attempted before, and while we do not think any method can claim to be the absolute best for the task, as the task is subjective, the method we propose gives distinctive stylization when compared to existing methods. On the request of the reviewers we also did a user study where we found that our method is competitive to the existing methods in terms of the trade-offs between stylization and content preservation as evaluated by human subjects. We are looking forward to any questions or queries the reviewers might have given our responses.

### Details and discussion of user study

We selected 20 random stylized videos from a combination of 15 content videos, 15 style images. We considered four other leading style transfer methods for comparison, namely AdaIN, MAST, MCCNet, AdaAttN.

We took a total of 900 votes from 15 users for this user study. We asked the users to vote separately for three preferences: Style, Content and Overall, i.e. which of the five presented options (i) transfers the style best, (ii) preserves the content best and (iii) is preferable overall.

The results of the user study are given in the attached PDF.

We observe that MCCNet performs best by a big margin in Style preference (31.3% MCCNet vs 20% AdaAttn and 16.7% proposed). While, our method leads in Content preference, closely followed by AdaAttN (43.3% vs 41%). In the case of Overall preference our method leads, and is closely followed by both AdaAttN, MCCNet (28.7%, 28.3%, 27.7% resp.). MCCNet tends to heavily stylize the output and distort the content, while the proposed method as well as AdaAttN maintain the content better. However, the styling as we observe in numerous qualitative examples in the supplementary as well, is quite different for these top performing methods. Hence we conclude that the proposed method is a competitive stylization method useful for end users.


### Discussion on the Optical flow metric table

As a response to a comment by reviewer qz5E, we could add two more methods for quantitative evaluation in the rebuttal period, i.e. SANET, MAST. We are giving the results here (Table in attached PDF) as they might be interesting for other reviewers too. Optical flow metric is one of the possible metrics for the task, it helps to make sure that the stylization method is preserving the motion in the original clip and there are no obvious/drastic failures. All the style transfer methods are expected to distort the content in some ways while performing stylization. Considering the average optical flow error our method has the least mean optical flow error and thus we can conclude our method is at par wrt the leading methods for the task. Our method also achieves mostly rank 1 and 2 among the methods.

---

### Author Response · Authors · 2023-08-10
**Link to video comparison**

Please find the video on a comparison of the proposed method with a method using 2D CNNs for stylization, which used optical flow as well (response to qZ5E).

https://anonymfile.com/drzOX/comparison-with-2d-style-transfer-with-optical-flow.mp4
OR (Use the below link in case of frequent site redirection from the above link)
https://xfl.jp/FoMuB2

---

### Decision · Program_Chairs · 2023-09-21

**Decision:**

Accept (poster)

**Comment:**

This paper presents a method to perform video stylization with a network that uses 3D convolutions as well as a host of other design choices to obtain temporally consistent results. While reviewers were initially skeptical about the degree of novelty of this idea, during the discussion phase the authors managed to convince reviewers that simply using 2D CNNs for the task introduces artifacts. As a result, the reviewer consensus shifted to accept. Reviewers also appreciated that this work introduces a large scale dataset that will encourage follow up research in this domain.